# Magnetic Resonance Imaging Predictors of Surgical Difficulty in Transforaminal Endoscopic Lumbar Discectomy for Far-Lateral Disc Herniation Under Local Anesthesia

**DOI:** 10.3390/biomedicines13040778

**Published:** 2025-03-23

**Authors:** Yong Ahn, Sungsoo Bae, Dae-Jean Jo, Byung-Rhae Yoo

**Affiliations:** 1Department of Neurosurgery, Kyung Hee University Hospital at Gangdong, Kyung Hee University College of Medicine, Seoul 05278, Republic of Korea; apuzzo@hanmail.net; 2Department of Neurosurgery, Gil Medical Center, Gachon University College of Medicine, Incheon 21565, Republic of Korea; byungryoo@gilhospital.com

**Keywords:** endoscopic surgical procedures, extraforaminal lumbar disc herniation, foraminal stenosis, local anesthesia, lumbar vertebrae, magnetic resonance imaging, pain management

## Abstract

**Background/Objectives**: Transforaminal endoscopic lumbar discectomy (TELD) is a minimally invasive spinal surgery known for its effectiveness, lower complication rates, faster recovery, and ability to be performed under local anesthesia. However, foraminal narrowing or access pain during the transforaminal approach can delay or hinder surgery in patients with far-lateral lumbar disc herniation (LDH). The objectives of this study were to identify predictive factors from preoperative magnetic resonance imaging (MRI) findings and demographics and discuss the optimization of surgical strategies. **Methods**: This retrospective study included 75 patients with far-lateral LDH who underwent TELD. Preoperative demographics and MRI findings were analyzed. Surgical data, including operative time, length of hospital stay, and intraoperative pain, were recorded. Postoperative outcomes, including complications, revision surgeries, and global outcomes based on the modified Macnab criteria, were evaluated. Preoperative clinical and radiological factors affecting the operative data and results were analyzed. **Results**: A higher foraminal stenosis grade was significantly correlated with prolonged operative time (*p* < 0.01) and extended hospital stay (*p* < 0.01). Extraforaminal LDH was associated with more severe access pain (*p* < 0.01) owing to increased nerve root irritation. Access pain was significantly correlated with operative time (*p* < 0.01) and hospital stay (*p* < 0.01). Appropriate surgical techniques and intraoperative pain management can mitigate these challenges. **Conclusions**: Preoperative MRI findings, particularly the grade of foraminal narrowing and herniation zone, can predict surgical difficulty and outcomes in TELD for far-lateral LDH. These insights can guide tailored strategies to reduce access pain and improve procedural success under local anesthesia.

## 1. Introduction

Transforaminal endoscopic lumbar discectomy (TELD), a key procedure in minimally invasive spinal surgery, was effective in numerous randomized controlled trials (RCTs) [1,2,3,4] and meta-analyses [5,6,7,8,9,10,11,12,13]. Its advantages, such as a lower complication rate, preservation of normal tissues, minimal stab incision, and early recovery, make it an appealing surgical option for lumbar disc herniation (LDH) [14,15,16,17,18,19]. A unique feature of TELD is its ability to be performed under local anesthesia, which reduces the risks associated with general anesthesia and promotes quicker postoperative mobilization.

However, performing the procedure under local anesthesia through a narrow foramen to access the epidural space and disc can present significant surgical difficulties. These include access pain, endoscopic procedural pain, and prolonged operative times required for sufficient decompression, which have historically posed barriers to the widespread adoption of this effective minimally invasive surgical technique [20,21,22]. Furthermore, these problems can be exacerbated in cases of foraminal or far-lateral LDH combined with foraminal stenosis by intervening in the approach window near the exiting nerve root and dorsal root ganglion [23,24]. This can lead to surgical failure or incomplete procedures, compromising patient satisfaction. Precise transforaminal approach techniques that avoid access pain and meticulous surgical manipulation to prevent procedural pain are crucial for resolving these difficulties. Equally important is the identification of predictive factors from preoperative imaging studies, which can help surgeons anticipate potential problems and plan strategies to address them effectively. Magnetic resonance imaging (MRI) is widely recognized as the most potent imaging modality for diagnosing degenerative spinal diseases including LDH [25]. Although studies have been conducted on the radiological findings related to the recurrence of endoscopic spine surgery [26], research analyzing MRI-based prediction rules for surgical difficulty in endoscopic procedures performed under local anesthesia remains scarce.

The primary objective of this study was to investigate the factors influencing surgical difficulty and TELD data for far-lateral LDH performed under local anesthesia. In particular, we aimed to determine whether preoperative MRI findings could reliably predict intraoperative challenges and postoperative outcomes. The predictive potential of MRI findings can significantly enhance patient selection and preparation, thereby improving the overall TELD process and patient satisfaction. The results revealed that the grade of foraminal stenosis and herniation zone can predict surgical difficulty and outcomes in TELD for far-lateral LDH. These insights can guide tailored strategies to reduce access pain and improve procedural success under local anesthesia.

## 2. Materials and Methods

### 2.1. Study Design and Patient Selection

This retrospective study included patients diagnosed with far-lateral LDH who underwent TELD under local anesthesia at a single tertiary hospital between January 2018 and December 2021. The Institutional Review Board of Gachon University Gil Medical Center approved the study (GDIRB2023-210, 24 June 2023), and written informed consent was obtained from all participants. Patients were included if they presented unilateral radiculopathy refractory to conservative treatment for at least six weeks and had MRI-confirmed single-level far-lateral LDH that occurred lateral to the spinal canal covering the foraminal and/or extraforaminal region [23,24,27] (Figure 1). Patients with intracanalicular LDH confined to the spinal canal (central or subarticular LDH), previous lumbar surgery at the same level, spondylolisthesis exceeding Grade I, or severe spinal deformities were excluded.

### 2.2. Preoperative Assessment

Demographic data, including age, sex, body mass index (BMI), and duration of symptoms were collected. All patients underwent detailed clinical and imaging evaluations including MRI and plain radiography. Preoperative MRI was reviewed to assess key factors, including foraminal stenosis grade (grade 0–3), zone (foraminal vs. extraforaminal), degree (protruded vs. extruded), and LDH level. Baseline demographic data, including sex, age, BMI, and symptom duration, were also collected.

The degree of foraminal stenosis was assessed on sagittal T1- and T2-weighted MRI using a 4-point grading scale [28,29]. In this system, a grade of 0 indicated a normal neuroforamen without any abnormalities. Grade 1 corresponded to mild foraminal stenosis, characterized by perineural fat obliteration in two opposing directions. Grade 2 represented moderate foraminal stenosis with perineural fat obliteration in four directions. Grade 3 indicated severe foraminal stenosis accompanied by visible morphological changes in the nerve root (Figure 2).

### 2.3. Surgical Technique

Transforaminal endoscopic lumbar discectomy was performed using the transforaminal approach under local anesthesia combined with sedation as needed [16,17]. Premedication included intramuscular administration of midazolam (0.05 mg/kg) and intravenous administration of fentanyl (0.8 μg/kg) on call. Additional fentanyl doses were administered as required depending on the patient’s vital signs and sedation level. The patients were then placed in a prone position, with their knees flexed on a radiolucent spine table. The procedure comprised two processes, namely (1) a transforaminal percutaneous approach under fluoroscopic guidance and (2) selective discectomy under direct endoscopic visualization (Figure 3).

Under fluoroscopic guidance, an 18-G needle was percutaneously inserted posterolaterally at approximately a 45° angle. The needle was inserted into the disc through the foraminal window, and intraoperative discography was performed using a contrast medium and indigo carmine. A sequential dilation technique was then employed until the working sheath was docked at the foraminal zone, outside or inside the disc surface (Figure 3A).

An ellipsoidal working channel endoscope was introduced through the working sheath, initiating a thorough selective discectomy under endoscopic visualization. Herniated disc fragments and compressed neural tissues were visualized (Figure 3B) and distinguished using instrumental dissection with a probe, forceps, and a radiofrequency tip. After adequate dissection and release of the annular anchorage, the herniated disc fragments were selectively removed using endoscopic forceps (Figure 3C).

The final point of the procedure was determined by the free mobilization of the nerve root and strong pulsation of the dural sac after sufficient discectomy (Figure 3D).

### 2.4. Data Collection

Intraoperative parameters, including operative time, access pain during the transforaminal approach, and procedural pain during endoscopic decompression, were recorded. Access pain was defined as pain experienced during the transforaminal approach under fluoroscopic guidance. This pain may have been caused by irritation of the exiting nerve root when the approaching device passed through the foraminal window. Procedural pain was defined as the pain experienced during endoscopic discectomy and decompression under endoscopic visualization. The intensity of each pain during TELD was classified on a four-point scale according to a published article [22]: (1) minimal (no or negligible irritation response), (2) mild (mild but tolerable, visual analog scale [VAS] 1–3), (3) moderate (definitive complaint of pain, VAS 4–6), and (4) severe (screaming and twisting in pain, VAS > 6). Moderate or severe pain was considered “significant”.

Postoperative outcomes, including length of hospital stay and requirement for postoperative procedures, were recorded during the follow-up period. The modified Macnab criteria [30] were used to evaluate global outcomes.

### 2.5. Statistical Analysis

The effects of independent variables such as sex, age, BMI, LDH type, LDH zone, and foraminal stenosis grade on dependent variables such as operative time, hospital stay, access pain, procedural pain, and global outcomes were assessed using the following statistical methods.

A univariate analysis was conducted to evaluate the relationship between the independent categorical variables and dependent outcomes. For continuous dependent variables, a one-way analysis of variance was performed to compare the means across categorical groups. For categorical dependent variables, the Pearson chi-square test was used to assess associations with independent variables.

A multivariate analysis was also performed using the following methods. Operative time and hospital stay: Multivariate linear regression was performed to evaluate the relationship between continuous dependent variables and independent predictors. Adjusted R-squared values and *p*-values for the F-statistic were used to assess the overall fit of the model and significance. Access pain, procedural pain, and global outcome: Multinomial logistic regression was used to analyze categorical dependent variables with more than two levels. Pseudo R-squared values and likelihood ratio test *p*-values were used to evaluate model significance. Categorical variables, such as LDH type, LDH zone, and foraminal stenosis grade, were treated as factor variables using dummy encoding. A significance threshold of *p* < 0.05 was used to identify statistically significant results.

All analyses were performed using SPSS (version 22.0; IBM Corp., Armonk, NY, USA) and the Python Statsmodels package (Statsmodels v0.13.5; USA).

## 3. Results

### 3.1. Demographics and Clinical Outcome

The study included 75 patients with an average follow-up duration of 31.1 months (range: 12–64 months). Of these, 24 (32.0%) were men and 51 (68.0%) were women, with a mean age of 61.24 years (range, 21–83 years). The operative levels were as follows: L2–3 in 3 patients (4.0%); L3–4, 12 (16.0%); L4–5, 31 (41.3%); and L5–S1, 29 (38.7%). Regarding LDH zones, 53 patients (70.7%) had foraminal herniation, whereas 22 (29.3%) had extraforaminal herniation. A summary of preoperative demographics is presented in Table 1.

Intraoperative pain, assessed using a four-point scale for transforaminal access, was categorized as no or minimal pain in 46 patients (61.3%); mild pain in 18 (24.0%); moderate pain in 6 (8.0%); and severe pain in 5 (6.7%). Consequently, the proportion of patients who experienced significant pain (moderate to severe) during access was 14.7% (11 out of 75 patients). Similarly, endoscopic procedure pain was rated as minimal in 22 patients (29.3%); mild in 41 (54.7%); moderate in 8 (10.7%); and severe in 4 (5.3%), with 16.0% (12 out of 75 patients) classified as having significant procedure-related pain.

The mean operative time was 62.7 min (range: 30–120 min), and the mean hospital stay was 2.3 days (range: 1–9 days). Outcomes assessed using the modified Macnab criteria showed excellent results in 12 patients (16.0%); good results in 49 (65.3%); fair in 11 (14.7%); and poor in 3 (4.0%). Symptomatic improvement was noted in 96.0% of the patients, while the success rate (defined as good or excellent outcomes) was 81.3%. Postoperative dysesthesia occurred in six patients (8.0%) and was managed with oral medication combined with a nerve root block. No major complications, such as dural tears, infections, or hematomas, have been reported. Three patients with poor outcomes underwent open revision surgery during follow-up.

### 3.2. Univariate Analysis of Predictors of Surgical Difficulty

According to the univariate analysis, the foraminal stenosis grade had a significant impact on operative time and hospital stay. Patients with narrower foraminal windows tended to have prolonged operative time (*p* = 0.0017) and hospital stay (*p* = 0.003).

A significant association was observed between the LDH zone and access pain. Patients with extraforaminal LDH tended to experience more severe pain during the transforaminal approach (*p* = 0.002). Access pain also significantly influenced the operative time (*p* = 0.0004) and hospital stay (*p* = 0.026).

No other preoperative factors had a significant effect on complications or global outcomes.

### 3.3. Multivariate Analysis of Predictors of Surgical Difficulty

The results of the multivariate analysis were consistent with those of the univariate analysis. Operative time was analyzed using multivariable linear regression, which identified the foraminal stenosis grade as a statistically significant predictor. Patients with significant foraminal narrowing (grade 2 or 3) demonstrated an average increase in operative time of 21.77 min compared to that in those with minimal foraminal stenosis (grade 0 or 1) (β = 21.77, 95% CI: [6.65, 36.89], *p* = 0.005, Table 2). Hospital stay was also evaluated using linear regression, which demonstrated a significant effect of the foraminal stenosis grade on the length of stay. Severe foraminal stenosis (grade 3) was associated with an average increase of 1.16 days in hospital stay compared to that in less severe cases (β = 1.16, 95% CI: [0.35, 1.98], *p* = 0.006, Table 2).

Access pain was analyzed using multinomial logistic regression, which revealed that LDH zone was significantly associated with access pain. Patients with extraforaminal LDH exhibited a higher likelihood of experiencing access pain compared to those with foraminal LDH (β = 2.44, 95%CI: [0.80, 4.08], *p* = 0.003, Table 2).

The effects of access and procedural pain on operative time, hospital stay, and global outcomes were evaluated (Table 3). Multivariable linear regression analysis identified access pain as a statistically significant predictor of operative time (β = 21.78, 95%CI: [9.54, 34.02], *p* = 0.001) and hospital stay (β = 1.50, 95%CI: [0.40, 2.60], *p* = 0.008).

In summary, the foraminal window significantly influenced the operative time and hospital stay, with a narrower foraminal window leading to longer operative times and prolonged hospital stays. Similarly, extraforaminal LDH was a significant determinant of access pain, and a more extraforaminal location resulted in a more painful surgical approach. Additionally, increased access pain was associated with longer operative times and extended hospital stays, further emphasizing its impact on surgical outcomes. No significant predictors were identified for global outcomes based on the modified Macnab criteria or procedural pain.

## 4. Discussion

### 4.1. Prediction of Surgical Difficulty in TELD for Far-Lateral LDH

This study confirmed that the severity of foraminal narrowing correlates with increased operative time and extended hospital stay. Additionally, extraforaminal LDH, compared with foraminal LDH, is associated with more severe approach-related pain and may indirectly contribute to longer operative time and prolonged hospital stay. Preoperative MRI can help predict these challenges. By anticipating these factors, surgeons can refine their surgical approaches and anesthesia plans to achieve better outcomes.

Our data indicate that although foraminal window narrowing and the zone of LDH were strongly associated with surgical difficulty, including operative time and access pain, they did not correlate with global outcomes based on the modified Macnab criteria. Ultimately, no MRI findings or clinical factors were identified as predictors of the final clinical outcome. This suggests that, despite initial surgical challenges, patients can achieve favorable long-term recovery if adequate neural decompression is achieved. Further prospective studies are warranted to explore predictive factors that may influence long-term clinical outcomes. Additionally, among the three patients who had poor outcomes, all experienced symptom recurrence within six months and eventually required additional open surgery. However, their preoperative MRI findings and surgical difficulties were not statistically correlated with poor outcomes. This highlights the complexity of predicting long-term prognosis and suggests that factors beyond imaging and initial surgical challenges may contribute to poor outcomes. Further studies with a larger cohort are needed to identify potential predictors of suboptimal long-term results.

One of the key advantages of the transforaminal endoscopic approach is that it can be performed under local anesthesia. The benefits of local anesthesia include its minimally invasive nature, its suitability for medically compromised or older patients who may not tolerate general anesthesia, and the ability to receive immediate patient feedback during surgery, enabling real-time nerve protection. However, incorporating objective intraoperative neuromonitoring has recently been recommended to further enhance safety by continuously monitoring and preventing nerve irritation or injury during the procedure [31]. Although neuromonitoring was not utilized in this study, we acknowledge its role in improving surgical safety and enhancing the reliability of outcomes.

### 4.2. Impact of Foraminal Stenosis Grade on Perioperative Data

A higher grade of foraminal stenosis was strongly associated with a prolonged operative time and extended hospital stay. Although the underlying foraminal stenosis itself did not directly lead to a significant increase in access pain, the decompression process in these cases often required additional time because of the complexity of addressing stenotic pathology. Although previous studies have suggested that the grade of foraminal stenosis can serve as an indicator for determining the necessity of lumbar spinal surgery [32], there is a notable lack of research directly linking the grade of stenosis to the operative time or hospital stay in lumbar spinal surgery. In this study, we hypothesized that severe foraminal stenosis in decompression surgeries using the transforaminal approach may prolong the operative time because of the increased effort required to establish an adequate approach window and working space. This extended operative time may contribute to increased postoperative discomfort, ultimately leading to a longer hospital stay. These findings underscore the need for careful surgical planning and precise surgical execution in patients with severe foraminal stenosis to mitigate its impact on surgical efficiency and recovery.

### 4.3. Foraminal vs. Extraforaminal LDH: Impact on Surgical Complexity

In TELD for extraforaminal LDH, patients experienced greater access pain, longer operative times, and longer hospital stays than those with foraminal LDH. A narrower safe working zone during the transforaminal endoscopic approach increases the likelihood of exiting nerve root irritation and pain [20,21,22]. In the present study, disc herniation within the extraforaminal zone may have further reduced the safe working zone during the transforaminal approach, thereby prominently exposing the exiting nerve root along the access pathway. This increased exposure likely exacerbated access pain, which, in turn, contributed to prolonged operative times and hospital stays (Figure 4). Therefore, particular attention should be paid to the strategies that minimize nerve root irritation and access pain while performing TELD for extraforaminal LDH.

### 4.4. Inherent Difficulties of the Transforaminal Endoscopic Approach

The transforaminal endoscopic approach offered several advantages, including the ability to bypass multiple normal anatomical structures and directly access lumbar lesions without nerve retraction. This allowed for effective neural decompression under local anesthesia. However, the inherent challenges must be addressed to achieve optimal outcomes.

The first challenge is the risk of injuring the existing nerve root during entry through the foraminal window. A common mistake among novice endoscopic spine surgeons is the inadvertent stimulation of the exiting nerve root during access, which can result in severe access pain or, in worse cases, neurological deficits due to nerve injury.

The second challenge involves a narrow foraminal window and a relatively thick facet joint, which can obstruct or significantly delay effective access to the target lesion. These limitations are particularly pronounced in cases of far-lateral LDH, in which the anatomical constraints of the foraminal or extraforaminal zones exacerbate the difficulties of the transforaminal approach. Special attention should be paid to improving techniques and planning while performing TELD for far-lateral LDH to minimize complications and enhance surgical efficiency.

### 4.5. Strategies to Reduce Access Pain and Operative Time

The proper management of access pain and foraminal stenosis is critical for successful TELD. Access pain may prolong the operative time and hospital stay. Furthermore, it may lead to surgical failure or psychological trauma. Several strategies can be used to optimize the transforaminal approach and reduce pain.

First, an outside-to-inside approach is recommended for far-lateral LDH. Starting outside the foramen ensures that the surgical trajectory avoids excessive manipulation of sensitive and inflamed neural structures. Selecting an initial point as far as possible from the exiting nerve root minimizes irritation and provides safe access for working. The surgeon should gently tap the working sheath at the outer surface of the foramen and not into the foramen while monitoring the patient’s response. The leading edge of the working sheath should be directed away from the exiting nerve root. Second, preemptive nerve root blocks help reduce pain while maintaining neural feedback, allowing safer and more controlled decompression. Third, intraoperative pain management is crucial to relieve procedure-related pain. For patients experiencing pain during surgery, administering intravenous anesthetics (e.g., midazolam and fentanyl) or epidurals (lidocaine/ropivacaine) can effectively alleviate symptoms and facilitate procedural completion. Finally, precise and rapid foraminal widening is crucial for ensuring a safe working space and neural decompression. Resection of the superior articular process and synovial joint using endoscopic burrs is the primary procedure in endoscopic foraminotomy. A high-speed drill can be used safely and rapidly until the ligamentum flavum and epidural fat are exposed.

### 4.6. Limitations of the Study

Despite the inclusion of consecutive cases that were analyzed longitudinally, this study has certain limitations. First, as this was a retrospective study with a relatively small sample size, considerable inherent biases may have affected the interpretation of the findings. We acknowledge that a prospective or randomized controlled design would provide more reliable and generalizable results. Nevertheless, we consistently collected quantitative data on procedure-related pain and foraminal window measurements over time and employed appropriate statistical methods to derive meaningful conclusions. Second, access to pain and postoperative data may be affected by conditions other than numerical demographics or imaging findings such as patient sensitivity, anesthetic conditions, underlying pain status, and social conditions. Therefore, some bias may have been involved in the interpretation of the operative data. Third, the follow-up period was relatively short, and long-term outcome data were insufficient. However, the primary focus of this study was to analyze the factors influencing surgical difficulty and immediate postoperative data. While long-term outcomes warrant further investigation in future studies, we were able to obtain quantitative data to assess factors affecting perioperative and immediate results, providing meaningful insights despite the follow-up limitations. Fourth, the study relies solely on pain scales, which may reduce the reliability of the results. Incorporating other objective measures, such as functional outcome scores, imaging-based metrics, blood pressure, and pulse, could improve the credibility of the findings. However, the pain scale, including access pain and procedural pain, has been widely used in previous research [22]. Consequently, we utilized procedure-related pain and operative time as key variables for assessing surgical difficulty in TELD and were able to obtain meaningful results. Future studies should aim to incorporate a broader range of outcome measures to further enhance the quality of research in this field. Finally, variability in imaging interpretation and measurement could introduce potential inconsistencies, particularly in the classification of access pain. Future studies should include an agreement analysis to improve the reliability of these assessments.

## 5. Conclusions

This study highlights the significance of preoperative MRI predictors in assessing procedural challenges in TELD for far-lateral LDH, particularly with regard to intraoperative pain, operative time, and length of hospital stay. Patients with more extraforaminal herniations and smaller foraminal windows may experience surgical difficulties, leading to significant access pain and a prolonged operative time. Optimized surgical planning and tailored techniques may enhance procedural safety and efficiency and ultimately improve patient outcomes.

## Figures and Tables

**Figure 1 biomedicines-13-00778-f001:**
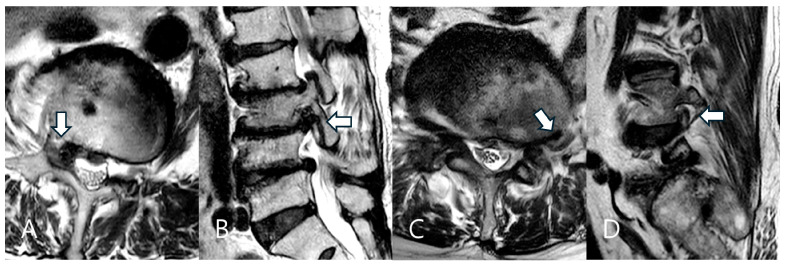
Two types of far-lateral lumbar disc herniation (LDH) based on the herniation zone. Foraminal LDH in a 75-year-old female patient at the right L3–L4 level (**A**,**B**). The herniated disc is located in the foraminal zone (arrow). Extraforaminal LDH in a 70-year-old male patient at the left L4–L5 level (**C**,**D**). The herniated disc is located in the extraforaminal zone (arrow).

**Figure 2 biomedicines-13-00778-f002:**
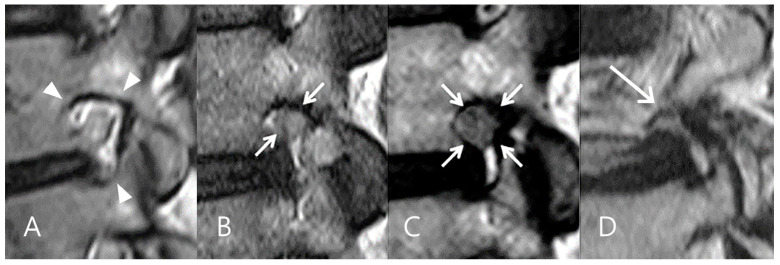
Grades of foraminal narrowing on sagittal MRI. (**A**) Grade 0 indicates a sufficient foraminal window space (arrowheads). (**B**) Grade 1 represents mild foraminal narrowing with perineural fat obliteration in two opposing directions (arrows). (**C**) Grade 2 represents moderate foraminal narrowing with perineural fat obliteration in all four directions (arrows) without morphologic changes in the nerve root. (**D**) Grade 3 indicates severe foraminal narrowing with morphologic changes in the nerve root (arrow).

**Figure 3 biomedicines-13-00778-f003:**
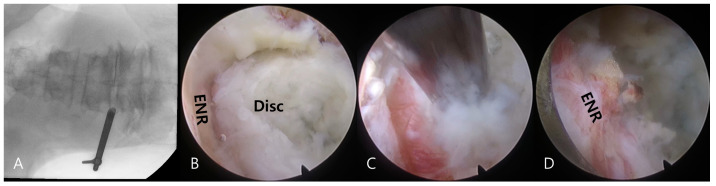
Intraoperative images during transforaminal endoscopic lumbar discectomy for a left extraforaminal lumbar disc herniation at the L4–5 level. Fluoroscopic images demonstrate the docking of the working sheath and the exploration of the foraminal and extraforaminal zones (**A**). Transforaminal selective discectomy is performed under endoscopic visualization. Note the herniated disc fragment (Disc) compressing the exiting nerve root (ENR) (**B**). The herniated disc fragment is removed using an endoscopic forceps (**C**). The final endoscopic view shows decompression of the ENR (**D**). The successful completion of the procedure is confirmed by free mobilization and a strong pulsation of the nerve root.

**Figure 4 biomedicines-13-00778-f004:**
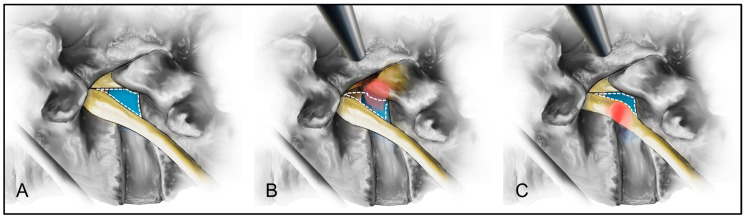
Schematic comparison of the safety working zones for the transforaminal approach in foraminal and extraforaminal lumbar disc herniation (LDH). The safety working zone (blue) in a standard lumbar disc is illustrated (**A**). For the foraminal LDH (red), the safety working zone (blue) may be narrowed but remain preserved for the transforaminal approach (**B**). In contrast, extraforaminal LDH (red) significantly narrows the safety working zone (blue), increasing irritation of the exiting nerve root and more significant access pain during the transforaminal approach (**C**).

**Table 1 biomedicines-13-00778-t001:** Preoperative demographics.

Items		Total (N = 75)	
Age (years), mean, SD		61.2	10.5
Sex	Male	24	32.0%
	Female	51	68.0%
BMI (kg/m^2^), mean, SD		23.7	3.5
Symptom Duration	Acute	8	10.7%
	Subacute	11	14.7%
	Chronic	56	74.6%
Level	L2–3	3	4.0%
	L3–4	12	16.0%
	L4–5	31	41.3%
	L5–S1	29	38.7%
Side	Left	39	52.0%
	Right	36	48.0%
LDH Type	Extruded	64	85.3%
	Protruded	9	12.0%
	Bulging, diffuse	2	2.7%
LDH Zone	Foraminal	53	70.7%
	Extraforaminal	22	29.3%
Grade (foraminal narrowing)	1	8	10.7%
	2	26	34.7%
	3	41	54.6%

BMI, body mass index; LDH, lumbar disc herniation; SD, standard deviation.

**Table 2 biomedicines-13-00778-t002:** Influence of preoperative predictors on surgical difficulties: results of regression and logistic regression analyses.

Items	Variables	Coefficient (β)	95% CI	*p*-Value	Model Fit
Operative time	FS grade	21.77	[6.65, 36.89]	0.005	R^2^ = 0.165
Hospital stay	FS grade	1.16	[0.35, 1.98]	0.006	R^2^ = 0.184
Access pain	Zone of LDH	2.44	[0.80, 4.08]	0.003	Rp^2^ = 0.2737
Procedure pain	No predictors	-	-	>0.05	Rp^2^ = 0.0776
Macnab	No predictors	-	-	>0.05	Rp^2^ = 0.1198

CI, confidence interval; FS, foraminal stenosis; LDH, lumbar disc herniation; Rp^2^, pseudo R-squared.

**Table 3 biomedicines-13-00778-t003:** Influence of intraoperative pain on clinical outcomes: results of regression and logistic regression analyses.

Items	Variables	Coefficient (β)	95% CI	*p*-Value	Model Fit
Operative time	Access pain	21.78	[9.54, 34.02]	0.001	R^2^ = 0.159
	Procedure pain	−1.29	[−13.11, 10.52]	0.828	
Hospital stay	Access pain	1.50	[0.40, 2.60]	0.008	R^2^ = 0.101
	Procedure pain	−0.88	[−1.94, 0.18]	0.101	
Macnab	Access pain	0.73	[−0.86, 2.32]	0.365	Rp^2^ = 0.0113
	Procedure pain	−0.43	[−2.21, 1.35]	0.637	

CI, confidence interval; Rp^2^, pseudo R-squared.

## Data Availability

The data presented in this study are available upon request from the corresponding author.

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
