# Peer review of "Magnetic Resonance Imaging Predictors of Surgical Difficulty in Transforaminal Endoscopic Lumbar Discectomy for Far-Lateral Disc Herniation Under Local Anesthesia"

_biomedicines, 2025, doi:10.3390/biomedicines13040778_

Round 1

Reviewer 1 Report

Comments and Suggestions for Authors

1. Three patients with poor outcomes, please discuss it.

2. please lable the blue and red in the picture in the Figure 4.

3. The foraminal stenosis grade and extraforaminal LDH can make the surgery more difficult. Why these factors do not impact the surgery outcome should be discuss in the article.

Author Response

Reviewer 1

  1. Three patients with poor outcomes, please discuss it.

Response: Thank you for your interest and insightful comments. We have added a discussion on the relationship between global clinical outcomes and preoperative or intraoperative factors in the Discussion section. Additionally, we have provided our perspective on the three patients who experienced poor outcomes.

“Additionally, among the three patients who had poor outcomes, all experienced symptom recurrence within six months and eventually required additional open surgery. However, their preoperative MRI findings and surgical difficulties were not statistically correlated with poor outcomes. This highlights the complexity of predicting long-term prognosis and suggests that factors beyond imaging and initial surgical challenges may contribute to poor outcomes. Further studies with a larger cohort are needed to identify potential predictors of suboptimal long-term results.”

  1. please label the blue and red in the picture in the Figure 4.

Response: The blue-colored zone represents the safety working zone, and the red-colored spot indicates the herniated fragment. As per your recommendation, we have revised the figure legend as follows:

Figure 4. Schematic comparison of the safety working zones for the transforaminal approach in foraminal and extraforaminal lumbar disc herniation (LDH). The safety working zone (blue) in a standard lumbar disc is illustrated (a). For the foraminal LDH (red), the safety working zone (blue) may be narrowed but remain preserved for the transforaminal approach (b). In contrast, extraforaminal LDH (red) significantly narrows the safety working zone (blue), increasing irritation of the exiting nerve root and more significant access pain during the transforaminal approach (c).”

  1. The foraminal stenosis grade and extraforaminal LDH can make the surgery more difficult. Why these factors do not impact the surgery outcome should be discuss in the article.

Response: Thank you for your constructive comment. As we demonstrated in Tables 2 and 3, the global outcomes based on the modified Macnab criteria were not related to foraminal window narrowing or the zone of LDH. We have added the following statement to the Discussion section:

“Our data indicate that although foraminal window narrowing and the zone of LDH were significantly associated with surgical difficulty, including operative time and access pain, they did not correlate with global outcomes based on the modified Macnab criteria. Ultimately, no MRI findings or clinical factors were identified as predictors of the final clinical outcome. This suggests that, despite initial surgical challenges, patients can achieve favorable long-term recovery if adequate neural decompression is achieved. Further prospective studies are warranted to explore predictive factors that may influence long-term clinical outcomes.”

Reviewer 2 Report

Comments and Suggestions for Authors

Major Revision Request for Manuscript

Dear Authors, 

We have thoroughly reviewed the manuscript submitted to the journal. While the study is about an important topic in neurosurgery, we have identified several significant limitations that must be addressed before the manuscript can be considered for publication. Below, we provide a detailed evaluation and suggestions for major revision.

The retrospective nature of the study inherently limits its robustness and objectivity. A prospective or randomized controlled design would provide more reliable and generalizable results.To add on, the sample size of 75 patients is inadequate to draw statistically significant conclusions. The reviewer strongly recommends expanding the sample size to enhance the study's statistical power.

The mean follow-up period of 31 months is relatively short and insufficient for assessing long-term outcomes and potential late complications. Extending the follow-up duration would provide a more comprehensive evaluation of the procedure’s efficacy and safety.

There is no control group fort his study. The absence of a control group is a major limitation of the study. Including a comparative control group, such as patients undergoing transforaminal endoscopic lumbar discectomy (TELD) under general anesthesia, would significantly strengthen the findings and allow for a better analysis of the procedure.

The study relies solely on subjective pain scales, which reduces the reliability of the results. We recommend incorporating objective measures, such as functional outcome scores or imaging-based metrics, blood pressure or pulse to improve the credibility of the findings.

The study does not bring substantial innovation to the field. Local anesthesia for endoscopic lumbar discectomy has been documented in early reports but was later largely abandoned due to nerve root irritation concerns. Discussing the historical use of local anesthesia, the reasons for its decline, and the rationale for revisiting it in this study would provide a more meaningful context. However, local anestesia is better for the patients and surgeons demand a less invasive method for surgery. Local anestesia is a better choice comparative to general anestesia.

The absence of neuromonitoring data is a methodological gap. Neuromonitoring is a standard practice to mitigate nerve root irritation, particularly in procedures performed under local anesthesia. Including neuromonitoring would improve the study’s methodology. Neuromonitoring is becoming a worldwide choice of surgeons performing lumbar surgeries.

The absence of postoperative imaging, particularly MRI data, limits the objective evaluation of surgical success. Adding this data is critical to assess outcomes such as decompression efficacy and residual pathology. Many surgeons demand a MRI scan after the surgery.

Addressing these points will enhance the quality and impact of the study, making it more suitable for publication. Please submit a revised manuscript along with a point-by-point response to these comments.

Author Response

Reviewer 2

Dear Authors, 

We have thoroughly reviewed the manuscript submitted to the journal. While the study is about an important topic in neurosurgery, we have identified several significant limitations that must be addressed before the manuscript can be considered for publication. Below, we provide a detailed evaluation and suggestions for major revision.

The retrospective nature of the study inherently limits its robustness and objectivity. A prospective or randomized controlled design would provide more reliable and generalizable results. To add on, the sample size of 75 patients is inadequate to draw statistically significant conclusions. The reviewer strongly recommends expanding the sample size to enhance the study's statistical power.

Response: We sincerely appreciate the reviewer’s constructive and thoughtful feedback. As acknowledged in the Limitation subsection of the Discussion section, this study is fundamentally a retrospective prognostic analysis, and we recognize that such a design inherently carries potential biases. However, we have employed the most appropriate statistical methods to analyze predictive factors for surgical difficulty, ensuring the rigor and reliability of our findings. Despite the limitations of sample size and study design, we believe that our results convey meaningful insights. We kindly ask the reviewer to consider these aspects in evaluating the study. We revised the Limitations of the Study subsection as follows:

“Despite the inclusion of consecutive cases that were analyzed longitudinally, this study has certain limitations. First, as this was a retrospective study with a relatively small sample size, considerable inherent biases may have affected the interpretation of the findings. We acknowledge that a prospective or randomized controlled design would provide more reliable and generalizable results. Nevertheless, we consistently collected quantitative data on procedure-related pain and foraminal window measurements over time and employed appropriate statistical methods to derive meaningful conclusions.”

The mean follow-up period of 31 months is relatively short and insufficient for assessing long-term outcomes and potential late complications. Extending the follow-up duration would provide a more comprehensive evaluation of the procedure’s efficacy and safety.

Response: We appreciate your insightful comment. In fact, our primary goal was finding predictors for perioperative surgical difficulty rather than long-term outcome including potential late complications. Despite the short follow-up period we could obtain quantitative data to analyze the factors to impact the perioperative or immediate results. We added the followings statement to the Limitations of the Study subsection.

“Third, the follow-up period was relatively short, and long-term outcome data was insufficient. However, the primary focus of this study was to analyze the factors influencing surgical difficulty and immediate postoperative data. While long-term outcomes warrant further investigation in future studies, we were able to obtain quantitative data to assess factors affecting perioperative and immediate results, providing meaningful insights despite the follow-up limitations.”

There is no control group for this study. The absence of a control group is a major limitation of the study. Including a comparative control group, such as patients undergoing transforaminal endoscopic lumbar discectomy (TELD) under general anesthesia, would significantly strengthen the findings and allow for a better analysis of the procedure.

Response: We agree with the reviewer’s opinion that the inclusion of a control group is an important factor in enhancing the quality and reliability of a study. However, the primary objective of this study was not to compare surgical outcomes or anesthesia methods but rather to analyze the factors determining surgical difficulty in TELD performed under local anesthesia. Despite the absence of a control group, we were able to derive meaningful findings. A comparative study on these predictors could be considered in future research, which would further strengthen the reliability of the results.

The study relies solely on subjective pain scales, which reduces the reliability of the results. We recommend incorporating objective measures, such as functional outcome scores or imaging-based metrics, blood pressure or pulse to improve the credibility of the findings.

Response: We appreciate the reviewer’s valuable insight. To date, there have been limited studies evaluating intraoperative or perioperative surgical difficulties in endoscopic spine surgery performed under local anesthesia. The factors suggested by the reviewer are indeed important measurements that could enhance the reliability of such evaluations. In this study, we primarily adopted procedure-related pain and operative time as key variables for assessing surgical difficulty, with the pain scale specifically based on prior literature [22]. As a result, we were able to derive meaningful findings from these parameters. In response to the reviewer’s suggestion, we have incorporated the following statement into the manuscript:

“Fourth, the study relies solely on pain scales, which may reduce the reliability of the results. Incorporating other objective measures, such as functional outcome scores, imaging-based metrics, blood pressure, and pulse, could improve the credibility of the findings. However, the pain scale, including access pain and procedural pain, has been widely used in previous literature [22]. Consequently, we utilized procedure-related pain and operative time as key variables for assessing surgical difficulty in TELD and were able to obtain meaningful results. Future studies should aim to incorporate a broader range of outcome measures to further enhance the quality of research in this field.”

We appreciate the reviewer’s thoughtful recommendations and will consider integrating additional objective measures in future research.

The study does not bring substantial innovation to the field. Local anesthesia for endoscopic lumbar discectomy has been documented in early reports but was later largely abandoned due to nerve root irritation concerns. Discussing the historical use of local anesthesia, the reasons for its decline, and the rationale for revisiting it in this study would provide a more meaningful context. However, local anesthesia is better for the patients and surgeons demand a less invasive method for surgery. Local anesthesia is a better choice comparative to general anesthesia.

The absence of neuromonitoring data is a methodological gap. Neuromonitoring is a standard practice to mitigate nerve root irritation, particularly in procedures performed under local anesthesia. Including neuromonitoring would improve the study’s methodology. Neuromonitoring is becoming a worldwide choice of surgeons performing lumbar surgeries.

Response: We sincerely appreciate the reviewer’s insightful comments. One of the key advantages of the transforaminal endoscopic approach is that it can be performed under local anesthesia. The benefits of local anesthesia include: (1) it inherently represents a less invasive surgical technique, (2) it is particularly beneficial for medically compromised or older patients who may not tolerate general anesthesia well, and (3) it allows direct patient feedback during surgery, which serves as a natural form of neuromonitoring. While there are certain surgical challenges associated with this approach, requiring an appropriate learning curve, advancements in surgical techniques, and improvements in local anesthesia methods have contributed to favorable outcomes in recent reports [1-6]. We firmly believe that TELD under local anesthesia remains an effective option for appropriately selected patients. Establishing precise guidelines for anesthesia selection can help maximize the effectiveness of endoscopic spine surgery.

References

  1. Sairyo K, Egawa H, Matsuura T, Takahashi M, Higashino K, Sakai T, Suzue N, Hamada D, Goto T, Takata Y, Nishisho T, Goda Y, Sato R, Tsutsui T, Tonogai I, Kondo K, Tezuka F, Mineta K, Sugiura K, Takeuchi M, Dezawa A. State of the art: Transforaminal approach for percutaneous endoscopic lumbar discectomy under local anesthesia. J Med Invest. 2014;61(3-4):217-25. doi: 10.2152/jmi.61.217.
  2. Sanusi T, Davis J, Nicassio N, Malik I. Endoscopic lumbar discectomy under local anesthesia may be an alternative to microdiscectomy: A single centre's experience using the far lateral approach. Clin Neurol Neurosurg. 2015 Dec;139:324-7. doi: 10.1016/j.clineuro.2015.11.001.
  3. Yoshinari H, Tezuka F, Yamashita K, Manabe H, Hayashi F, Ishihama Y, Sugiura K, Takata Y, Sakai T, Maeda T, Sairyo K. Transforaminal full-endoscopic lumbar discectomy under local anesthesia in awake and aware conditions: the inside-out and outside-in techniques. Curr Rev Musculoskelet Med. 2019 Jun 24;12(3):311-317. doi: 10.1007/s12178-019-09565-3.
  4. Yang L, Pan YL, Liu CZ, Guo DX, Zhao X. A retrospective comparative study of local anesthesia only and local anesthesia with sedation for percutaneous endoscopic lumbar discectomy. Sci Rep. 2022 May 6;12(1):7427. doi: 10.1038/s41598-022-11393-4.
  5. Chen Z, Wang X, Cui X, Zhang G, Xu J, Lian X. Transforaminal Versus Interlaminar Approach of Full-Endoscopic Lumbar Discectomy Under Local Anesthesia for L5/S1 Disc Herniation: A Randomized Controlled Trial. Pain Physician. 2022 Nov;25(8):E1191-E1198.
  6. Mooney J, Laskay N, Erickson N, Salehani A, Mahavadi A, Ilyas A, Mainali B, Nowak B, Godzik J. General vs Local Anesthesia for Percutaneous Endoscopic Lumbar Discectomy (PELD): A Systematic Review and Meta-Analysis. Global Spine J. 2023 Jul;13(6):1671-1688. doi: 10.1177/21925682221147868.

In response to the reviewer’s suggestion, we have added the following statement to the Discussion section:

“One of the key advantages of the transforaminal endoscopic approach is that it can be performed under local anesthesia. The benefits of local anesthesia include its minimally invasive nature, its suitability for medically compromised or older patients who may not tolerate general anesthesia, and the ability to receive immediate patient feedback during surgery, enabling real-time nerve protection. However, incorporating objective intraoperative neuromonitoring has recently been recommended to further enhance safety by continuously monitoring and preventing nerve irritation or injury during the procedure [31]. Although neuromonitoring was not utilized in this study, we acknowledge its role in improving surgical safety and enhancing the reliability of outcomes.”

We appreciate the reviewer’s valuable suggestions and recognize the importance of intraoperative neuromonitoring in further optimizing patient safety during endoscopic spine surgery.

The absence of postoperative imaging, particularly MRI data, limits the objective evaluation of surgical success. Adding this data is critical to assess outcomes such as decompression efficacy and residual pathology. Many surgeons demand a MRI scan after the surgery.

Response: We agree with the reviewer’s perspective. Postoperative MRI plays a valuable role in enhancing the credibility of surgical outcomes and providing surgeons with critical feedback on the effectiveness of decompression. However, not all patients consent to postoperative MRI due to concerns about additional costs, and routine imaging may not always be feasible. At our institution, postoperative MRI is not performed as a standard protocol after TELD; instead, it is reserved for cases where clinical concerns arise or when specifically requested by the patient. We appreciate the reviewer’s understanding of these considerations.

Round 2

Reviewer 1 Report

Comments and Suggestions for Authors

Thanks for you reply. I think the readers will be interested in this article.

Reviewer 2 Report

Comments and Suggestions for Authors

Dear Authors,

Following a thorough review of the your responses and the revised manuscript entitled "Magnetic Resonance Imaging Predictors of Surgical Difficulty in Transforaminal Endoscopic Lumbar Discectomy for Far-Lateral Disc Herniation Under Local Anesthesia," the reviewer offers the following evaluation.

The authors have demonstrated a significant effort in addressing the concerns raised during the initial review process. Their responses were comprehensive and understanding of the study's limitations. The revisions implemented have notably enhanced the clarity of the manuscript. 

The authors have responded to our comments with well-reasoned explanations.While the study's inherent limitations remain, the authors have strengthened its clarity and focus. The reviewers overall assesment is upon addressing these points, the manuscript is suitable for publication.